# The Effect of Data Encoding on Relation Triplet Identification

**Steinunn Rut Friðriksdóttir**
Department of Computer Science
University of Iceland
Iceland
srf2@hi.is

**Hafsteinn Einarsson**
Department of Computer Science
University of Iceland
Iceland
hafsteinne@hi.is

## Abstract

This paper presents a novel method for creating relation extraction data for low-resource languages. Relation extraction (RE) is a task in natural language processing that involves identifying and extracting meaningful relationships between entities in text. Despite the increasing need to extract relationships from unstructured text, the limited availability of annotated data in low-resource languages presents a significant challenge to the development of high-quality relation extraction models. Our method leverages existing methods for high-resource languages to create training data for low-resource languages. The proposed method is simple, efficient and has the potential to significantly improve the performance of relation extraction models for low-resource languages, making it a promising avenue for future research.

## 1 Introduction

Relation extraction (RE) is a task in the field of natural language processing, aimed at identifying and extracting semantically meaningful relationships between entities present in text. A relation is generally extracted as an ordered triple $(E_1, R, E_2)$ where $E_1$ and $E_2$ refer to the entity identifiers and $R$ refers to the relation type. This task holds great significance in several practical applications, including information retrieval, knowledge management, and question-answering systems, among others (for a recent review, see (Yan et al., 2021)). The increasing availability of unstructured text data on the web has only served to underline the importance of relation extraction, as there is a pressing need to convert this data into structured information that can be eas-

ily accessed and analyzed. This challenge is especially pertinent for low-resource languages like Icelandic, where the limited availability of annotated data presents a significant impediment to the development of high-quality relation extraction models.

Relation extraction methods for English have evolved over the years, with early methods relying on hand-crafted rules, patterns, and statistical analyses (Soderland et al., 1995; Carlson et al., 2010; Kambhatla, 2004; Jiang and Zhai, 2007). With the advent of deep learning and the availability of large annotated corpora, more sophisticated methods have emerged (Liu et al., 2013; Xu et al., 2016; dos Santos et al., 2015). Deep learning models have shown promising results in extracting relationships between entities, outperforming traditional methods. Current state-of-the-art model, REBEL (Cabot and Navigli, 2021), performs joint relation extraction[1]. The progress in relation extraction for English has demonstrated the potential for using advanced techniques to extract meaningful relationships from large amounts of text data.

The challenge of developing effective RE methods for languages like Icelandic lies in the scarcity of annotated data. The performance of machine learning models heavily relies on the availability of large amounts of annotated training data. The limited availability of annotated text in low-resource languages creates a major challenge for training high-quality relation extraction models. To overcome this challenge, one way is to study methods to efficiently create training data based on existing methods for English. Our main question is whether models from high-resource languages can be used to efficiently create training and testing data for low-resource languages.

In this paper, we present a novel method for efficiently creating relation extraction data for low-

---

[1]That is, entity extraction and relation extraction are not two separate processes.

resource languages like Icelandic. Our method is based on replacing entities in the text with unique identifiers, translating the text to a high-resource language using machine translation, and then replacing the entities back in the translated text. Finally, we perform relation extraction on the translated text to obtain the relationships between entities. Our method is simple and only requires the location of entities in the text and a machine translation model. This approach leverages the availability of specialized models in high-resource languages to create training data for low-resource languages, thereby addressing the challenge posed by the scarcity of annotated data. Our method has the potential to significantly improve the performance of relation extraction models for low-resource languages, making it a promising avenue for future research.

## 2 Purpose

In a previous paper, we proposed a way of bootstrapping RE training data for low-resource languages (LRL) using a combination of machine translation and open RE methods (Friðriksdóttir et al., 2022). By automatically translating the LRL data into English, we were able to feed it directly into the high-resource language SOTA model before translating the relation triplets back to the LRL where it can serve as training data for a new LRL model.

While this method showed potential, the resulting data was jumbled by errors in translation. Some examples of this include people's names being perturbed by the multilingual translation model (resulting in *Alfreð* being changed into *Alfredo*, *Sveinn* into *Sweene* etc.), entities getting directly translated and thereby loosing their meaning (such as when the Icelandic name *Erlendur* gets directly translated as *foreign*) and unfortunate translation mishaps (such as when *Dauðarósir*, a novel by the Icelandic author Arnaldur Indriðason, gets translated as *Deathly Hallows*, a real novel by a different author).

In this paper, we hypothesize that these translation errors can be ameliorated by encoding the entities within the data before it gets translated, and then decoding them before they get sent into the high-resource RE model. Whereas the proposed method remains the same, this extra step in preprocessing the input data should result in more accurate predictions made by the RE model, which in turn makes for better training data.

## 3 Previous Work

Machine-translation has previously been used to create cross-lingual named entity recognition (NER) datasets, which led to improvement in NER for several languages (Dandapat and Way, 2016; Jain et al., 2019). In these earlier works, the text was translated directly, without any modifications, and the entities in the resulting text were matched heuristically to the entities in the untranslated text using word alignment methods. This works well for entities that translate correctly or change little in the translation process but can be limited by the translation system, specifically if the system translates entities incorrectly.

For RE in low-resource languages, there has been limited focus on building training and testing data efficiently. However, crosslingual transfer methods have been applied to improve RE models, such as using multilingual BERT (Nag et al., 2021). Universal dependencies and sequence-to-sequence approaches have also been employed for RE in low-resource languages (Taghizadeh and Faili, 2022). Finally, recent sequence-to-sequence approaches for English have focused on extracting both relations and relation types (Cabot and Navigli, 2021).

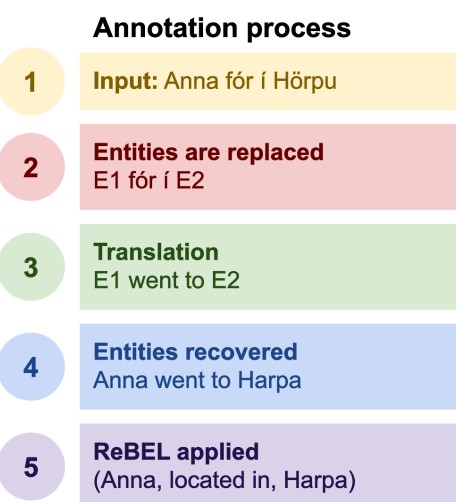

**Annotation process**

| | |
|---|---|
| 1 | **Input:** Anna fór í Hörpu |
| 2 | **Entities are replaced** E1 fór í E2 |
| 3 | **Translation** E1 went to E2 |
| 4 | **Entities recovered** Anna went to Harpa |
| 5 | **ReBEL applied** (Anna, located in, Harpa) |

Figure 1: Summary of the annotation process for relation extraction studied in this paper.

## 4 Methodology

A general overview of the annotation approach is demonstrated in Figure 1. Below, we outline the

methods used in the process.

## 4.1 Models

For translation, we use the model from Facebook AI's WMT21 submission (Tran et al., 2021). The model is multilingual and has well-performing Icelandic to English translation capabilities.

For relation extraction, we use REBEL (Cabot and Navigli, 2021) a sequence-to-sequence relation extraction model that achieves a micro-F1 score of 75.4 on the well-studied CONLL4 corpus (Roth and Yih, 2004) and 93.4 on the NYT corpus (Riedel et al., 2010).

## 4.2 Data

For evaluating the precision of the relation extraction method, we use the first 200 sentences of each category from the MIM-GOLD-EL corpus (Friðriksdóttir et al., 2022). The number of unique and lemmatized entities for each category are shown in Table 5 in the Appendix.

We also performed a further evaluation on 200 sentences chosen uniformly at random from the sentences annotated above to get an estimate of the precision, recall and F1-score on the evaluated text for encoded vs. not encoded entities. We used strict evaluation as in (Taillé et al., 2020), and we additionally required that the name of the entity perfectly matched the lemmatized version of the entity's name in the knowledge base. We would like to emphasize that this annotation task is cognitively more demanding than solely estimating precision as no gold data exists for relation extraction in Icelandic. The task requires the annotator to find all relations in a given text, instead of just labelling the output of a model as correct/incorrect. As REBEL is an open relation extraction model and thus accounts for a very large amount of different relations, we restricted ourselves to those that had already appeared in our data (a total of 145 types).

## 4.3 Encoding Entities

We use gold-annotated entities. Each entity receives its own identifier and is replaced by it. The first occurring entity in the text receives the identifier E0, the next one E1, et cetera. For the sake of clarity, we note that if an entity appears multiple times in the text, it receives the same entity for all occurrences. This makes it clear which encoding refers to which original entity when decoded.

## 5 Results

Our results indicate that using the encoding method proposed can increase the number of correctly identified relations between two entities by up to 9.7% (Table 1). It is evident that a higher number of relation triplets is proposed by REBEL when the data is not encoded (Table 2) and we suspect this is due to the text being more fluently English, i.e. it contains less foreign (in our case, Icelandic) words within the English translation which should make it more natural for the monolingually English RE model. On the other hand, having a higher number of relation triplets proposed introduces significantly more noise, making the percentage of correctly identified relation triplets in fact lower. Additionally, encoding the data reduces the number of translation inconsistencies and errors. Conjugation of nouns tends to be mangled by the translation model, creating examples such as *Steingríms [genitive] Davíðsson [nominative]*, accents get dropped (Ása becomes Asa) and Icelandic letters modified (Þorgrímur becomes Thorgrímur). Summary statistics can be seen in Table 4 and by category in Table 5 in the Appendix.

It's worth noting that the text categories from our corpora that contain news have the highest precision scores and improvements using our method. This is not surprising since they were translated using WMT21's newsdomain parameter which should make them better translated. All categories score higher in correctly identified relation triplets when using the encoding method except one, the adjudications category. This is likely because there are few entities reported in that category and most of them are anonymized, and not counted as a relation because an anonymous person A being related to an anonymous person B is not informative for a knowledge base.

When looking at the overall performance of our model in Table 3 we observe higher recall than precision, which is explained by the high number of relations reported by the model. Having a higher preference for recall than precision is of great importance in a labelling task such as this one since it is generally cognitively less demanding to label the output of the model as correct/incorrect rather than to identify the missing relations in the text. However, the task becomes longer with lower precision since most of the reported relations will be irrelevant. When using

encoded entities we observed a big difference for relation triplets where both items in the relation consisted of entities, 11.5% F1-score for encoded entities vs. 1.9% for non-encoding. When evaluating the correctness of all relations (i.e., not only those between gold labelled entities), we saw a slight drop in F1-scores using our encoding method. This is not surprising, since the encoding method is intended to improve the extraction of relations between gold-annotated entities, which could come at the cost of performance for extracting other relations.

| Category | Not | Encoded |
|---|---|---|
| Adjudications | 3.4% | 3.1% |
| Blog | 1.0% | 1.6% |
| Books | 1.5% | 5.3% |
| Emails | 0.3% | 5.5% |
| Newspaper 1 | 1.6% | 8.9% |
| Newspaper 2 | 1.5% | 7.1% |
| Laws | 0.0% | 0.5% |
| Radio/TV news | 0.3% | 7.6% |
| School essays | 2.7% | 4.0% |
| Scienceweb | 1.3% | 6.4% |
| Webmedia | 2.1% | 11.8% |
| Websites | 1.3% | 9.6% |
| Written to be spoken | 1.9% | 7.9% |

Table 1: Precision score per category for all relations between two established entities. The reported numbers are the percentage of relation triplets labelled as correct per text category within our corpus.

| | Not | Encoded |
|---|---|---|
| Total relations extracted | 8910 | 8870 |
| Entities in translation | 3476 | 4368 |
| Correct relations | 2450 | 2078 |
| Correct with entities | 135 | 541 |

Table 2: Aggregated results on the data in Table 1. Entities in translation refers to the number of times that entities from MIM-GOLD-EL appear lemmatized in the resulting translation. Correct relations refers to the total number of relations labelled as correct, regardless of whether or not they contain established entities. Correct with entities refers to relation triplets that contain established entities as both head and tail.

## 5.1 Qualitative Evaluation of Errors

We note that while technically correct, the machine translation model tends to give various different translations to a single entity which certainly influences the higher number of unique relation triplets proposed by the RE model when working with data that has not been encoded. For instance, the Icelandic political party Samfylkingin gets translated in four different ways (Confederation, Alliance, Social democrats and Social democratic party) as well as the rescue worker association Landsbjörg (translated as either accident insurance company, accident prevention association, emergency rescue association or accident prevention society). Encoding the data avoids the problem of having to backtrack the translations in order to figure out whether or not they refer to the same original entity.

Using the encoding method, we can additionally ensure that all extracted relation triplets contain entities that include the entire, lemmatized mention of the entity. As per Icelandic naming conventions, a person is generally only referred to by their full name the first time they are mentioned in a given text and afterwards only referred to by their first name. When working with data that has not been encoded, we therefore get relation triplets that include only the first name of a person, potentially conjugated, while the encoded data always ensures that the person's entire, lemmatized name is present within the triple. This creates more consistency and avoids ambiguity in the output data.

It should, however, be noted that REBEL itself occasionally jumbles entity mentions itself even though the data has been encoded. Examples of this include when REBEL proposes that a person is a part of a family by that person's last name (i.e. *(Ingibjörg Sólrún Gísladóttir, is a member of, Gísladóttir)* but this is not how things work in Icelandic where patronyms are used instead of traditional last names. Another example is when REBEL adds international endings to websites that already include their Icelandic endings (such as when *tonlist.is* becomes *tonlist.is.com*).

## 6 Discussion

In this work, our focus was on evaluating an encoding method that can lead to improved automated relation extraction in text such as Icelandic. To eliminate any errors due to the recognition of entities, we based this evaluation on gold-

| Method | Precision | Recall | F1-score | Evaluation |
|---|---|---|---|---|
| Encoded entities | 26.1% | 50.8% | 34.5% | All relations |
| | 6.3% | 65.2% | 11.5% | Between two entities |
| No encoding | 26.9% | 62.0% | 37.5% | All relations |
| | 1.0% | 30.4% | 1.9% | Between two entities |

Table 3: Precision, recall and F1-scores for a subset of 200 examples chosen at random to estimate false negatives and hence recall and F1-score.

annotated entities. However, for labelling relations, our method can be combined with existing NER models such as from (Snæbjarnarson et al., 2022) that achieves an accuracy of 98.8% for Icelandic. We further believe that it would be interesting to study our approach for relation extraction methods that only report possible relations between entities instead of all possible relations in the text, i.e., where the entities can be provided as input to the relation extraction model. For further study of efficiency, it could be interesting to compare this method to heuristics that match entities in translated text.

One limitation of our study is the strict evaluation approach. We deemed the output of REBEL to be incorrect if the entities were not in their lemmatized form, shortened or otherwise modified. For example, when talking about someone using their first name only, REBEL does not have sufficient context to disambiguate the entity for insertion into a knowledge-base. This could be addressed by first disambiguating the entities before the text is processed in this manner. We did not evaluate how much the performance could increase, but we believe that a good disambiguation model could have a significant effect on the result.

Our approach addresses the low-performance of modern machine translation systems in translating entities correctly. Therefore, we would expect that improvements in machine translation would make our approach obsolete. However, that would require better translations of named entities. Unfortunately, for Icelandic, we do not have a corpus of entities translated to other languages such as English. Transliteration of named entities is the process of translating entities across languages and has been performed for English and several other languages (Grundkiewicz and Heafield, 2018). Transliterating named entities could be an approach to improve machine translation for Icelandic and would possibly make it more reliable to

translate without any modifications to the source text and use word alignment to match entities between the source text and the translated text.

For creating data on relation extraction, we use machine-translation as an aid. However, to build a good relation extraction system for Icelandic, it might not be necessary to fine-tune the system on Icelandic. As an example, multilingual QA systems have shown good performance on Icelandic although they were not fine-tuned in QA for the language (Snæbjarnarson and Einarsson, 2022). We expect to see similar results for Icelandic and the data from this work can serve as a test set to measure the performance.

## 7 Conclusion

In conclusion, the proposed encoding method shows great potential for LRL, improving the percentage of correctly identified relations between entities by up to 9.7% for various categories of text. The method is simple and does not require any additional cost, making it ideal for languages where data is scarce and budget is limited. We note that this method can only be as good as the quality of the machine translation models as well as the RE methods for higher resourced languages. However, the encoding avoids several issues introduced by the bootstrapping method, making it more efficient with minimal effort.

## Acknowledgments

This research was conducted with funding from The Strategic Research and Development Programme for Language Technology. We thank the anonymous reviewers for providing helpful comments on this manuscript and Valdimar Ágúst Eggertsson for helpful discussions.

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

## A  Summary Statistics

To highlight the scope of this work, we list important summary statistics in Table 4 and by category in Table 5.

|                          | Encoded | Not Encoded |
|--------------------------|---------|-------------|
| Total relations extracted | 8870    | 8910        |
| Unique triplets          | 1558    | 1805        |
| Unique relation types    | 128     | 141         |

Table 4: Results on the overall data. Total relations refers to the total number of relations retrieved by REBEL. Unique triplets refer to the total number of unique relation triplets.

| Category        | # Unique Entities | # Not encoded | # Encoded |
|-----------------|-------------------|---------------|-----------|
| Adjudications   | 50                | 731           | 734       |
| Blog            | 75                | 689           | 757       |
| Books           | 118               | 752           | 740       |
| Emails          | 89                | 355           | 343       |
| Newspaper 1     | 207               | 696           | 676       |
| Newspaper 2     | 213               | 722           | 690       |
| Laws            | 96                | 694           | 651       |
| Radio/TV news   | 132               | 708           | 686       |
| School essays   | 89                | 747           | 753       |
| Scienceweb      | 180               | 707           | 702       |
| Webmedia        | 202               | 712           | 729       |
| Websites        | 184               | 699           | 678       |
| Written to be sp. | 184             | 728           | 731       |

Table 5: The first column shows the number of unique, lemmatized entities in the first 200 sentences of each category. The second column depicts the number of relations where the data has not been encoded and the third the number of relations where the data has been encoded.