# OpenReview forum: "The Effect of Data Encoding on Relation Triplet Identification"
_NoDaLiDa/2023/Conference — NoDaLiDa 2023_

### Official Review · Reviewer_cnBc · 2023-03-08
**Review of NoDaLiDa submission "The Effect of Data Encoding on Relation Triplet Identification"**

**Rating:** 7
**Confidence:** 4

**Review:**

This paper describes a clever shortcut to handle semantic analysis for low-resource languages: translate text into a high-resource language, perform analysis while retaining linkage to original mentioned entities. This method is of obvious utility for the given task and there might be other tasks which translate as well. The authors explore how preserving the authority of entities can be improved through judicious encoding of them.

The reader will be interested in error analyses and comparisons of how different language pairs and different tasks perform. This selected task is presumably more or less universal in that every human language encodes relations of some kind and that they are presumably somewhat syntax-independent. Where are the boundaries of usefulness for this approach? How much hinges on these two selected languages being quite similar and being both historically and culturally related (this is something that the examples in the paper discuss wrt naming conventions)? What errors happen and how can they be mitigated?

**Paper Type:**

Short paper

---

### Official Review · Reviewer_b9eo · 2023-03-08
**Well-written paper with a clear contribution, extending a current methodology for training RE models for low-resource languages.**

**Rating:** 9
**Confidence:** 4

**Review:**

Very well-written paper - it was a joy to read - with a clear contribution that expands on methodology for training RE models for low-resource languages.

*Pros*
- Small, focused contributions, that is a perfect fit for a short paper
- Well-written
- Clear methodology and related works

*Cons*
- None that I can identify

*Comments*

- What is your method for evaluating the translated triplets (every element in the "not" model and relation in "encoded"? By aligned token-span from the translation?
- You compare the _number of identified entities_ and the _precision_ of the "not" and "encoded" models:
> "It is evident that a higher number of relation triplets is proposed by REBEL when the data is not encoded (Table 2) and we suspect this is due to the text being more fluently English, i.e. it contains less foreign (in our case, Icelandic) words within the English translation which should make it more natural for the monolingually English RE model.  On the other hand, having a higher number of relation triplets proposed introduces significantly more noise, making the percentage of correctly identified relation triplets in fact lower"
  - Including metrics f1 measures, precision & recall, instead of accuracy might help you elaborate on this. Also, this might be able to shed light on the "books" result.

- Do you thoughts on why the model performs worse within the "books" category?

- Did you consider the length of the introduced foreign entities and relation and how that affect the model's ability to identify them?

- As many of the problems that you discuss relates to introducing foreign entities, would you benefit from a model that is not end-to-end, but that can be provided with the input entities? Then the identification of foreign entities would not be a problem, but maybe some other problems arise?


**Paper Type:**

Short paper

---

### Official Review · Reviewer_Uqef · 2023-03-09
**The paper needs further considerations with regard to the experiment setup, and evaluation**

**Rating:** 5
**Confidence:** 3

**Review:**

The paper introduces a corpus creation method for Icelandic tuple relation extraction based on an automatic translation between Icelandic and English. The study extends the previous work on cross-lingual annotation transfer for relation tuple extraction by replacing named entities in the source language with symbols that can be reverted after processing in the target language. The method performs promisingly in relatively limited experiments the authors have conducted.

However, it needs to be clarified how the entities are recognized on the source side. Do the authors train a separate NER model for Icelandic or rely on gold annotation? If the former, then the authors need to examine the effect of error propagation made by the NER system. If the latter, the authors need to propose a solution for applying their model to raw Icelandic text without annotation. Furthermore, the limited evaluation of the model does not tell us enough about the model's performance. It would be more helpful if the authors had reported the standard precision/recall/f-score in Table 1 and performed a detailed quantitative analysis on the transliterations (or other heuristics) of the named entities mentioned against the symbolic translation used in this study.

**Paper Type:**

Short paper

---

### Decision · Program_Chairs · 2023-03-17

Accept